# Structure-guided design of VAR2CSA-based immunogens and a cocktail strategy for a placental malaria vaccine

Rui Ma[1], Nichole D. Salinas[1], Sachy Orr-Gonzalez[2], Brandi Richardson[2], Tarik Ouahes[2], Holly Torano[2], Bethany J. Jenkins[3], Thayne H. Dickey[1], Jillian Neal[2], Junhui Duan[2], Robert D. Morrison[2], Apostolos G. Gittis[4], Justin Y. A. Doritchamou[3], Irfan Zaidi[2], Lynn E. Lambert[2], Patrick E. Duffy[2,3], Niraj H. Tolia [1]*

1 Host-Pathogen Interactions and Structural Vaccinology Section, Laboratory of Malaria Immunology and Vaccinology, National Institute of Allergy and Infectious Diseases, National Institutes of Health, Bethesda, Maryland, United States of America, 2 Vaccine Development Unit, Laboratory of Malaria Immunology and Vaccinology, National Institute of Allergy and Infectious Diseases, National Institutes of Health, Bethesda, Maryland, United States of America, 3 Pathogenesis and Immunity Section, Laboratory of Malaria Immunology and Vaccinology, National Institute of Allergy and Infectious Diseases, National Institutes of Health, Bethesda, Maryland, United States of America, 4 Structural Biology Section, Research Technologies Branch, National Institute of Allergy and Infectious Diseases, National Institutes of Health, Bethesda, Maryland, United States of America

* niraj.tolia@nih.gov

**Data Availability Statement:** All data needed to evaluate the conclusions in the paper are present in the paper and/or Supplementary Materials. Plasmids can be provided by N.H.T. pending

## Abstract

Placental accumulation of *Plasmodium falciparum* infected erythrocytes results in maternal anemia, low birth weight, and pregnancy loss. The parasite protein VAR2CSA facilitates the accumulation of infected erythrocytes in the placenta through interaction with the host receptor chondroitin sulfate A (CSA). Antibodies that prevent the VAR2CSA-CSA interaction correlate with protection from placental malaria, and VAR2CSA is a high-priority placental malaria vaccine antigen. Here, structure-guided design leveraging the full-length structures of VAR2CSA produced a stable immunogen that retains the critical conserved functional elements of VAR2CSA. The design expressed with a six-fold greater yield than the full-length protein and elicited antibodies that prevent adhesion of infected erythrocytes to CSA. The reduced size and adaptability of the designed immunogen enable efficient production of multiple variants of VAR2CSA for use in a cocktail vaccination strategy to increase the breadth of protection. These designs form strong foundations for the development of potent broadly protective placental malaria vaccines.

## Author summary

Malaria remains a major cause of death globally despite considerable dedication to eradication. *Plasmodium falciparum* is the most lethal malaria parasite that infects human. One reason for the parasite's deadly impact lies in the expression of the *Plasmodium falciparum* erythrocyte membrane protein 1 (PfEMP1) protein family. These proteins adhere to host receptors leading to parasite accumulation in various tissues and organs. VAR2CSA is a

scientific review and a completed material transfer agreement. Requests should be submitted to Richard Shimp (rshimp@niaid.nih.gov).

**Funding:** This work was supported by the Intramural Research Program of the Division of Intramural Research, National Institute of Allergy and Infectious Diseases (NIAID), National Institutes of Health (NIH) to N.H.T. The funders had no role in study design, data collection and analysis, decision to publish, or preparation of the manuscript.

**Competing interests:** NHT, RM and PED are inventors on a patent application related to this work. JYAD is an inventor on patent US9855321B2.

PfEMP1 protein that binds to a host receptor in the placenta causing severe outcomes for pregnant women and their unborn children including maternal anemia, low birth weight, still birth, and loss of pregnancy. The development of a vaccine to combat these devastating outcomes has long been anticipated, but the intricate and diverse nature of VAR2CSA has posed significant hurdles. Here, we adopted structure-guided principles to design a next-generation VAR2CSA vaccine candidate, HPISVpmv1, that is stable, has high production yields, and consists of the entire functional domains of VAR2CSA. Preclinical studies showed that HPISVpmv1 is immunogenic and effective, and the adoption of a cocktail immunization strategy produced antibodies that worked against multiple strains. This work provides new insights into vaccine development for malaria and other highly variable human pathogens.

## Introduction

Malaria during pregnancy is a major global health problem. The accumulation of *Plasmodium falciparum* parasites in the placenta of pregnant women causes severe outcomes for both the mothers and their offspring, resulting in high rates of maternal anemia, low birth weight, still-birth, and pregnancy loss [1–3]. Up to 200,000 infant deaths and 50,000 maternal deaths annually are attributed to malaria infection in pregnancy globally [4–6]. A body of evidence has shown that women naturally acquire resistance to placental malaria over successive pregnancies, indicating that a vaccine that replicates this immunity should protect against placental malaria [7,8].

VAR2CSA is a ~310 kDa *Plasmodium falciparum* erythrocyte membrane protein 1 (PfEMP1) family member protein. VAR2CSA is the leading vaccine candidate for placental malaria because it is expressed on the surface of infected red blood cells (iRBCs) and directly mediates the sequestration of iRBCs by binding to the placental receptor chondroitin sulfate A (CSA) [9,10]. Recently, the structures of VAR2CSA alone [11–13] and in complex with CSA [11,13] have been solved. The ectodomain of VAR2CSA consists of an N-terminal sequence (NTS), six or seven Duffy-binding-like (DBL) domains and three interdomain regions (IDs) [11–14]. Plasmodium parasites utilize a variety of DBL domain-containing proteins [15,16] to bind to host receptors [17,18]. The structural and mechanistic basis [19] for some of these interactions in erythrocyte invasion [20–27], cytoadherence [28,29] and placental sequestration [11–13] have been defined. Insights from antibody neutralization of parasite proteins [30–37] have enabled vaccine design for some antigens [38–41]. Antigens have the propensity to elicit both neutralizing human antibodies that protect against disease and interfering human antibodies that interfere with the function of neutralizing antibodies through an immune evasion mechanism called antigenic diversion [42]. Structure-guided design of VAR2CSA-based immunogens leveraging the available structural, mechanistic, and immune evasion mechanisms may improve placental malaria vaccines.

The overall architecture of VAR2CSA resembles the number 7, with a core structure comprising the segment from the NTS domain to ID3 and an arm structure composed of DBL5 and 6 [11] (Fig 1A and 1B). A 12-residue sugar segment of the placental receptor CSA binds within a channel formed by the NTS, DBL1, DBL2 and DBL4 domains [11,13] (Fig 1B and 1C). A potential second CSA binding site is located between DBL2 and ID2a [11] (Fig 1B and 1C). The cryo-EM structure indicates that the CSA binding sites are surrounded by highly polymorphic residues, and this variation hampers the development of a strain-transcending vaccine [11]. PRIMVAC and PAMVAC are two placental malaria vaccine candidates that are

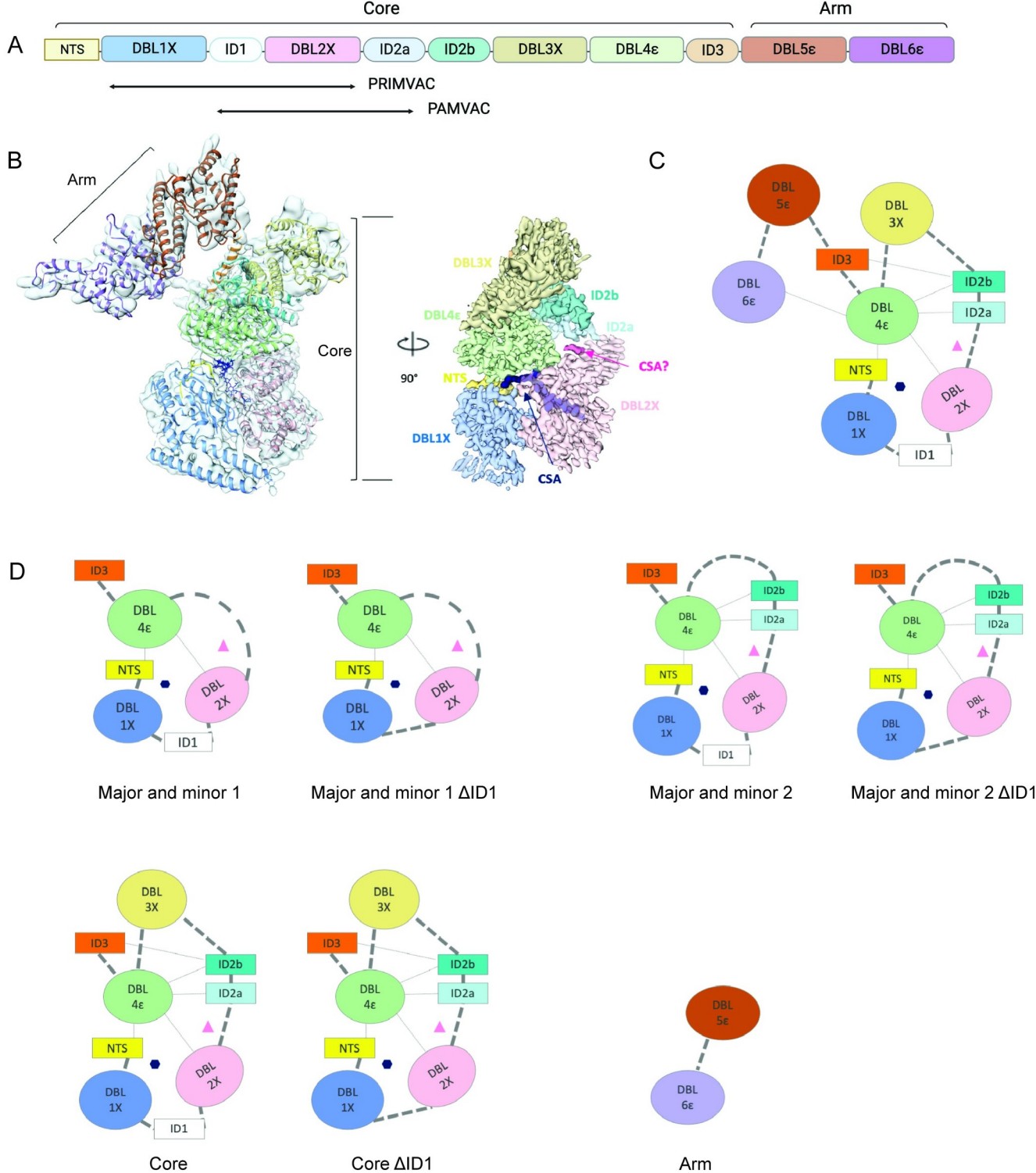

**Fig 1. Structure-guided design of VAR2CSA-based immunogens. (A**) Domain structures of the full-length VAR2CSA protein. (**B**) Left: VAR2CSA model fitted into the reconstructed density. Right: the Core structure is shown in surface with 70% transparency. The CSA 12-mer that binds within the major CSA binding channel is shown in dark blue, and the potential CSA monomer that binds within a putative minor CSA binding channel is shown in magenta. (**C**) Schematic drawing of the VAR2CSA structure and the major and minor CSA binding channels. The major binding channel and minor binding channel are represented by the dark-blue hexagon and magenta triangle, respectively. (**D**) Schematic of the designed immunogens. This figure was generated with the help of Biorender (https://www.biorender.com/) and PRISM 9.

in clinical trials that comprise part of the CSA binding channel [43–45]. The CSA binding sites do not include the immunodominant DBL3, DBL5 and DBL6 domains, and human antibodies preferentially target these domains [11,46,47].

Here, we used structure-based design to develop a VAR2CSA immunogen termed HPISVpmv1 that retains the conserved functional elements of VAR2CSA and eliminates the immunodominant polymorphic segments. Serology studies indicated that HPISVpmv1 elicited antibody titers equal to or greater than the full-length VAR2CSA ectodomain control after a two-vaccination regimen in rats. These antibodies showed potent inhibition of infected erythrocyte binding to CSA. These studies were performed using an adjuvant suitable for translation to clinical studies and form a strong foundation for the further development of these candidates as vaccines. Strain-transcending protection for placental malaria is hindered by the high sequence variation in VAR2CSA among parasite strains and isolates. We leveraged the reduced size and enhanced yield of HPISVpmv1 to expedite the production of five variants for a cocktail immunization strategy that provided broad protection against multiple strains.

## Results

### Structure-based design and production of VAR2CSA immunogens

We recently established the structural basis and binding mechanism for VAR2CSA engagement of CSA through three cryoEM structures of VAR2CSA alone and in complex with CSA [11]. Seven structure-based designs of VAR2CSA were developed based on these data (Figs 1A–1D and S1). The major and minor CSA binding channels of VAR2CSA are formed predominantly by the NTS, DBL1, DBL2 and DBL4 domains, and three immunogens (Major and minor 1, Major and minor 2 and Core) were developed that contain these critical domains (Figs 1D and S1). ID1 is located near the major CSA binding channel and is not observed in any of the cryoEM structures, likely due to inherent structural flexibility [11], and three additional immunogens where the ID1 domain was removed (ΔID1) were developed (Figs 1D and S1). Last, a construct that comprises DBL5 and DBL6 known as the arm was designed [11]. The arm is not involved in CSA binding [11,13] and serves as an excellent control for further study. This led to seven designs and the full-length ectodomain comparator.

All designs were expressed and purified from Expi293 cells with high purity and homogeneity as established by size-exclusion chromatography (SEC) and SDS PAGE analysis (S2A and S2B Fig). The yield of the immunogens varied from 1.3 mg/L to 260 mg/L (S2A Fig). The ΔID1 versions have consistently lower yields than the versions that include ID1, which indicates that ID1 might play a role in structural stability (S2A Fig). This is consistent with the finding that ID1 interacts with DBL2 and DBL4 [48] and might play a role in stabilizing the CSA binding channel. We analyzed the circular dichroism (CD) spectra of the immunogens, and the results indicate that all the immunogens have formed structures similar to the full-length VAR2CSA (S3 Fig).

### The designs are immunogenic and key designs elicit responses comparable to the full-length protein

Preclinical animal studies in rats are an established system to evaluate placental malaria vaccine candidates [49–51]. Rats were immunized with the designed immunogens and Complete Freund's Adjuvant (CFA)/Incomplete Freund's Adjuvant (IFA) three times to initially evaluate immunogenicity (Fig 2A). All designs elicited high antibody titers by ELISA after three vaccinations (Fig 2B). We evaluated the functional antibody response using the binding inhibition assay (BIA) with *P. falciparum* NF54 infected erythrocytes and pooled purified IgGs at 1 mg/

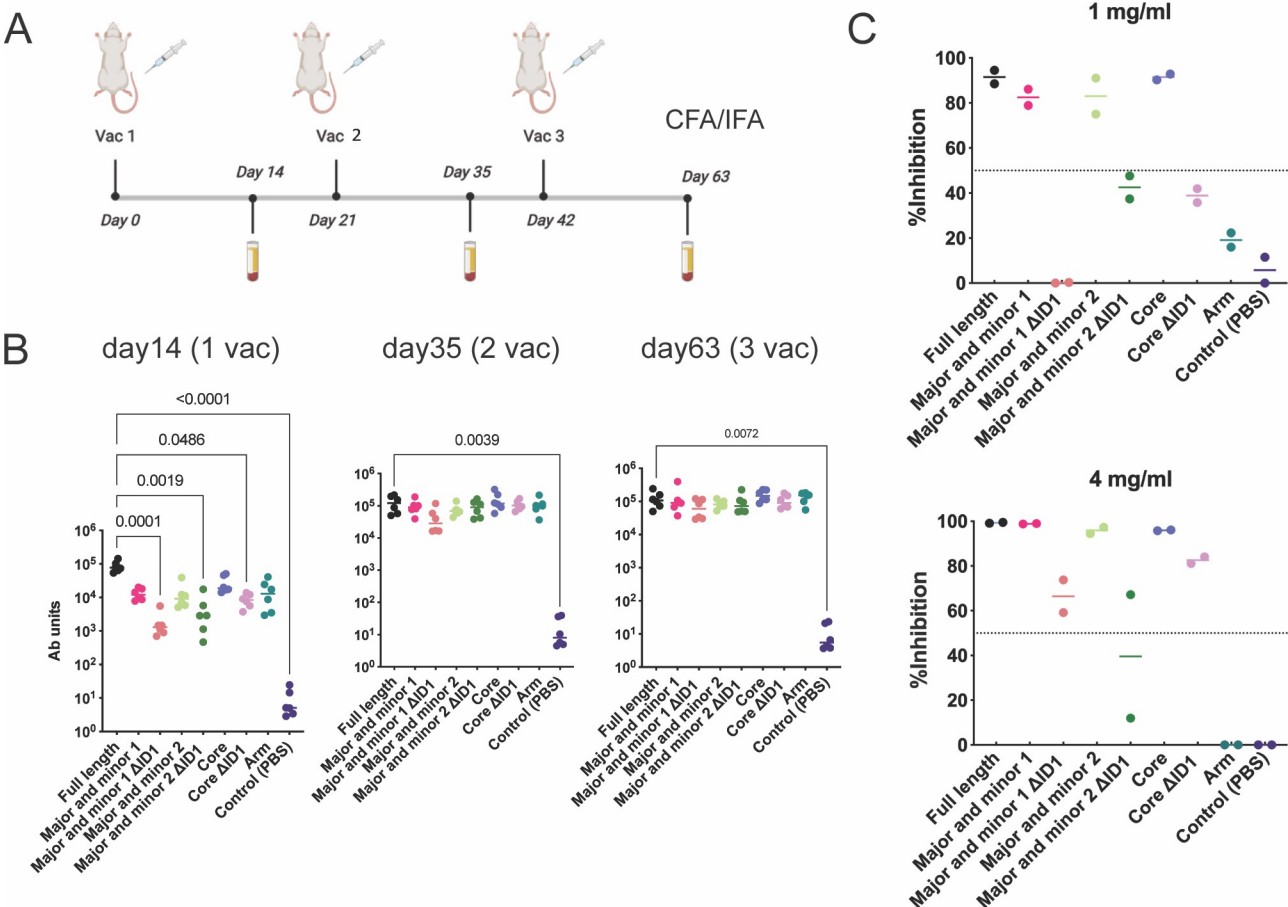

**Fig 2. The designed immunogens formulated in CFA/IFA elicit strong antibody responses and binding inhibitory activity.** (**A**) Schematic of the immunization schedule. (**B**) Antibody titers to full-length VAR2CSA NF54 measured two weeks after each vaccination by ELISA. Each point is the mean of three technical replicates for each individual rat, and the bar represents the median value for the group. The p-values were calculated using a two-sided Kruskal-Wallis non-parametric test with Dunn's multiple comparisons compared to the full-length protein and comparisons with p-values less than or equal to 0.05 are shown. (**C**) Purified pooled IgGs from serum after three vaccinations for each group were evaluated by BIA at 1 mg/ml (upper) or 4 mg/ml (bottom). The dashed line indicates the 50% inhibition level used as a cutoff for inhibitory or non-inhibitory activity. The data shown are derived from two independent experiments, each point is the mean of two technical replicates, and the bar represents the median value. This figure was generated with the help of Biorender (https://www.biorender.com/) and PRISM 9.

ml and 4 mg/ml. Major and minor 1, Major and minor 2, and Core elicit inhibitory activity (>70%) comparable to the full-length protein at both 1 mg/ml and 4 mg/ml IgG concentrations, while the arm did not (<30% inhibitory activity) (Fig 2C). In contrast, the ΔID1 versions of these designs showed less than 50% inhibitory activity at 1 mg/ml, although their inhibitory activity increased to greater than 50% at 4 mg/ml (Fig 2C). The reduced activity of the ΔID1 versions suggests that elimination of ID1 destabilizes the antigens and is consistent with the reduced expression and yield of the ΔID1 designs. These data indicate that Major and minor 1, Major and minor 2, and Core designs are promising candidate antigens. Of these three designs, Major and minor 1 is the most promising design, as it had the highest expression yields, approximately six-fold greater than the full-length protein, and elicited strong inhibitory activity (Figs 2B, 2C and 3A). We further optimized the construct's C-terminus based on the different definition of the boundaries of ID3 [11–13] to evaluate whether the C-terminal residues affected expression (S4A–S4C Fig). The shorter C-terminal construct resulted in the

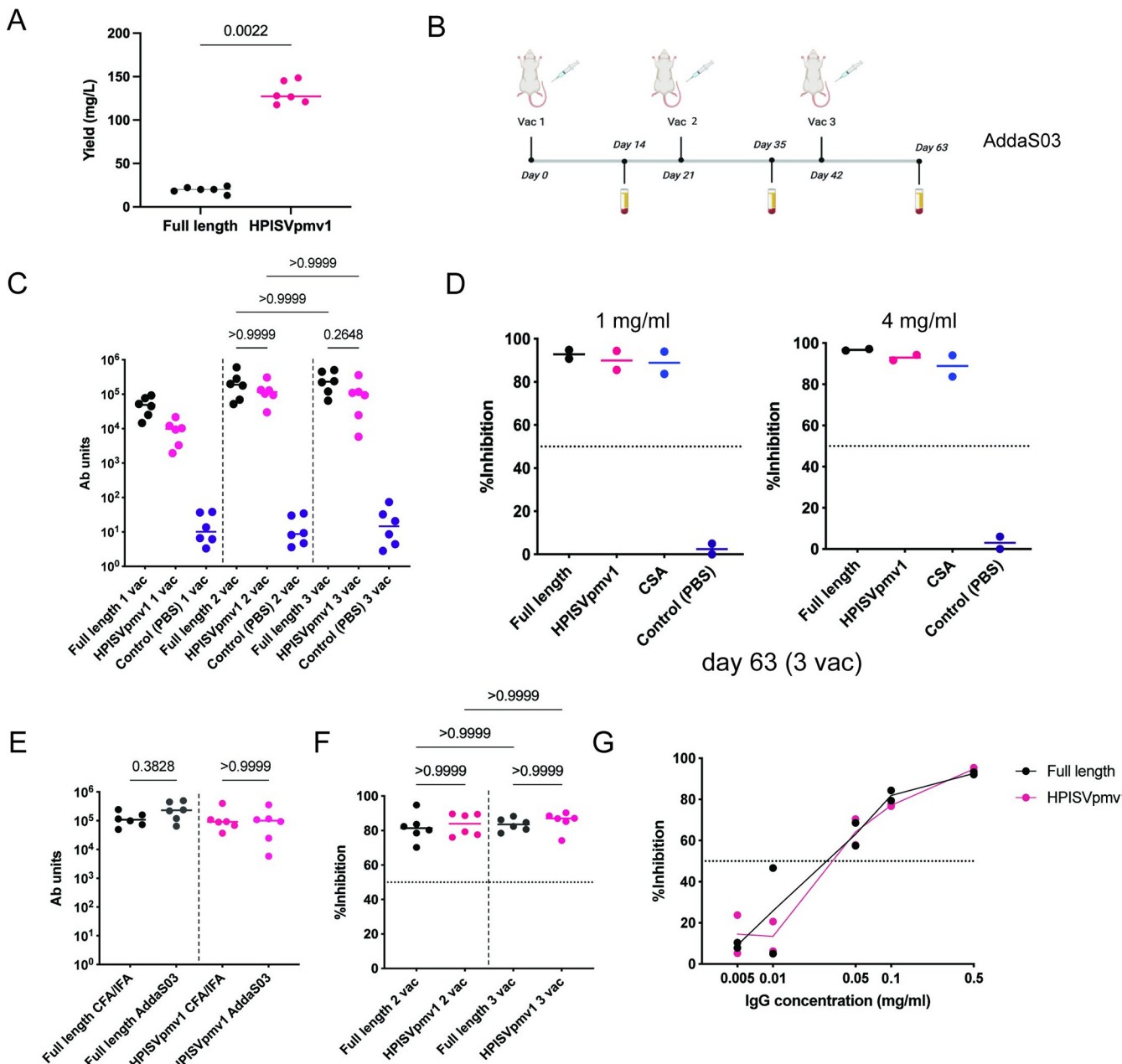

**Fig 3. HPISVpmv1 formulated in AddaS03 produced a strong antibody response and binding inhibitory activity upon vaccination of rats.** (**A**) Expression yield comparison between full length VAR2CSA and HPISVpmv1. The p-value of a Mann-Whitney test was shown. (**B**) Schematic of the immunization schedule. (**C**) Antibody titers measured two weeks after each vaccination by ELISA to full-length VAR2CSA NF54. Each point is the mean of three technical replicates for each individual rat, and the bar represents the median value for the group. A two-sided Kruskal-Wallis non-parametric statistical analysis followed by Dunn's test to correct for multiple comparisons of antibody titers elicited by full-length VAR2CSA and HPISVpmv1 showed no significant difference between day 35 (2 vac) and day 63 (3 vac), indicating that a two-vaccination regimen is sufficient. (**D**) Purified pooled IgGs from serum after three vaccinations for each group were evaluated by BIA at 1 mg/ml (left) or 4 mg/ml (right). The dashed line indicates the 50% inhibition level used as a cutoff for inhibitory or non-inhibitory activity. The data shown are derived from two independent experiments, and each point is the mean of two technical replicates. The bar represents the median value, and 0.1 mg/ml CSA was used as the positive control. (**E**) A two-sided Kruskal-Wallis non-parametric statistical analysis followed by Dunn's test to correct for multiple comparisons of antibody titers elicited by the two adjuvants used in this study at day 63 showed no significant difference, indicating that AddaS03 provides a strong immune response. (**F**) Purified IgGs from each individual rat were examined by BIA at 1 mg/ml for the full-length and HPISVpmv1 groups adjuvanted in AddaS03, demonstrating strong inhibitory activity in all animals. Two-sided Kruskal-Wallis non-parametric tests followed by Dunn's test to correct for multiple comparisons were performed to evaluate whether the inhibitory activity differed between the number of vaccinations, and no significant difference was observed. The 50% inhibition level used as a cutoff for inhibitory or non-inhibitory activity is highlighted as a dashed line. (**G**) Dose-dependent inhibition of the purified pooled IgG after two vaccinations from the full-length and HPISVpmv1 groups by BIA. The 50%

inhibition level used as a cutoff for inhibitory or non-inhibitory activity is highlighted as a dashed line. This figure was generated with the help of Biorender (https://www.biorender.com/) and PRISM 9.

highest expression levels, and the final optimized design was renamed HPISVpmv1 for simplicity.

CFA/IFA is a readily available research adjuvant that we employed for initial evaluation. However, it is imperative to evaluate the immune response when formulated in an adjuvant that can readily be translated to use in humans. The adjuvant AddaS03 is a research-grade adjuvant equivalent of AS03 that is suitable for use in humans [52–54]. The optimized HPISVpmv1 was subsequently moved forward for vaccination formulated in AddaS3 (Fig 3B). Serology studies indicated that HPISVpmv1 elicited a strong antibody response when formulated in AddaS03 compared to the PBS control (Fig 3C). The antibody titers elicited by HPISVpmv1 showed no significant difference when compared to the antibody titers elicited by the full-length protein after three vaccinations (Fig 3C). Furthermore, vaccination with HPISVpmv1/AddaS03 resulted in strong functional binding inhibitory activity that was indistinguishable from that of the full-length protein (Fig 3D). Finally, no differences in antibody titers between HPISVpmv1 formulated in either CFA-IFA or AddaS03 were observed (Fig 3E), indicating that AddaS03 is suitable for further development and that the results derived here will likely be applicable to the vaccination of humans with HPISVpmv1 formulated in AS03.

## Two vaccinations of HPISVpmv1 are sufficient to elicit inhibitory response

We examined whether two vaccinations would be sufficient to generate a peak functional immune response by comparing the antibody titers and binding inhibitory activity with antibodies purified from individual animals (Fig 3C). A two-vaccination regimen induced antibody titers that showed no difference from the three-vaccination regimen (Fig 3C). BIA results also indicate that the two-vaccination regimen is sufficient to elicit potent inhibitory activity (Fig 3F). Finally, we titrated the pooled purified antibodies after two vaccinations in the binding inhibition assay, which demonstrated potent functional activity as low as 0.05 mg/ml of purified antibodies consistent with the maximal inhibition afforded by the full-length protein (Fig 3G). These data indicate that a two-vaccination regimen may be suitable for further vaccine development.

## A cocktail vaccination strategy expands the breadth of protection

VAR2CSA vaccines typically have not achieved a strain-transcending immune response due to the highly polymorphic nature of VAR2CSA [14]. As with other VAR2CSA vaccines, the HPISVpmv1 NF54 design showed strong functional activity against *P. falciparum* NF54 infected cells but lacked activity against *P. falciparum* CS2 (containing a FCR3 VAR2CSA sequence) or WF12 (containing a 7G8 VAR2CSA sequence) infected cells [55] (Figs 3D, S5A and S5B). Similarly, full-length VAR2CSA, Core, Major and minor 2 induced homologous inhibitory activity but did not demonstrate strain-transcending activity (Figs 3D, S5A and S5B). These data indicate that the functional antibody response is largely strain specific.

Cocktail vaccination strategies that include several representative epitopes/immunogens have been shown to increase the breadth of protection for HIV in primates [56], influenza in mice [56,57], pneumococcal and COVID-19 in humans [58,59]. The HPISVpmv1 design has an increased yield, ease of production, ease of adaptation to diverse isolates and induces functional activity equivalent to the full-length protein. We therefore examined whether a cocktail approach could increase the breadth of protection using the structure-based design. We

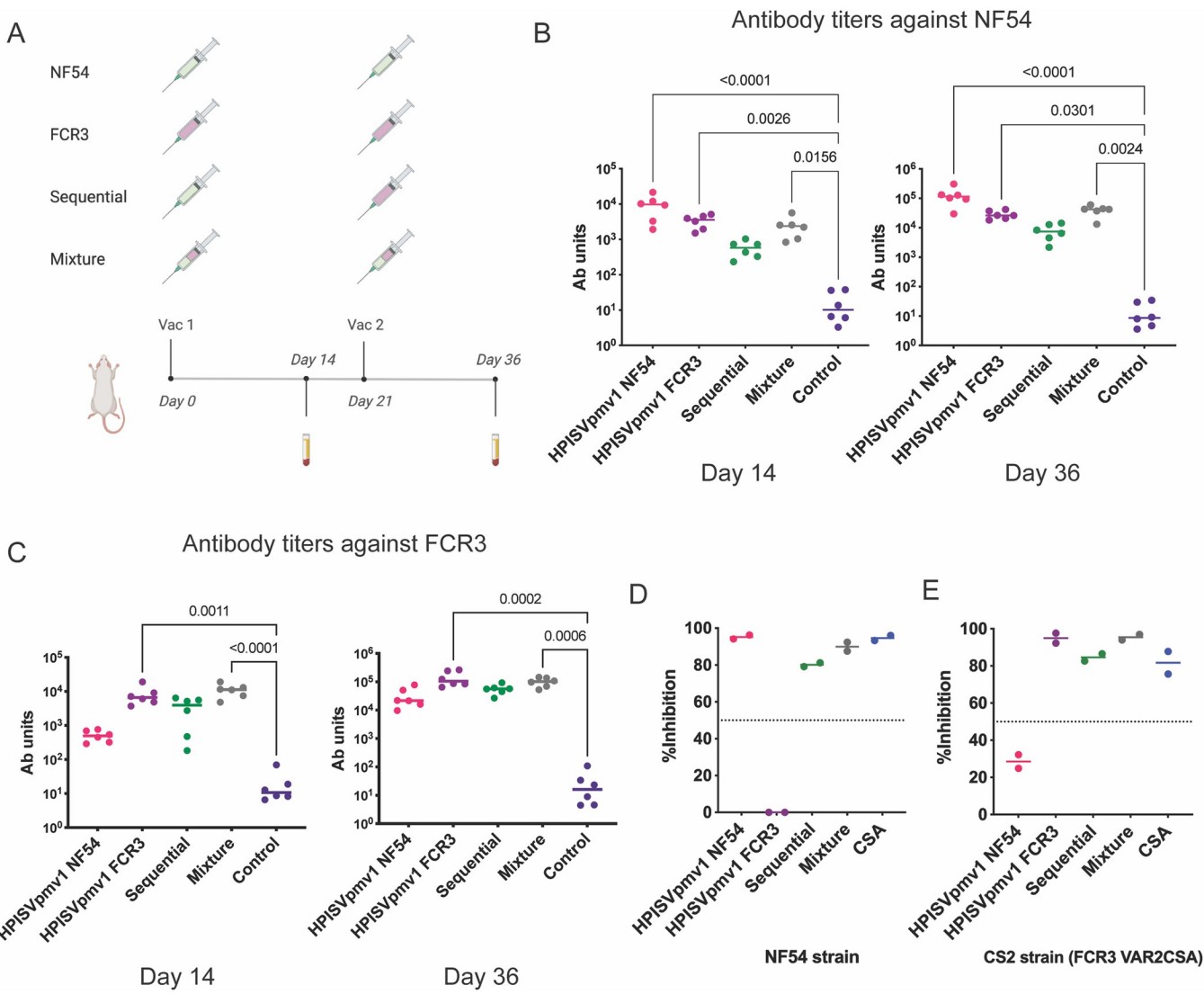

**Fig 4. Cocktail immunization with two variants increased the breadth of protection.** (**A**) The two-vaccination immunization schedule with the cocktail strategy illustrated above. The NF54 group was immunized with HPISVpmv1 based on the VAR2CSA NF54 sequence; the FCR3 group was immunized with HPISVpmv1 based on the VAR2CSA FCR3 sequence. The sequential strategy group was sequentially immunized first with HPISVpmv1 from NF54 and then boosted with HPISVpmv1 from FCR3. The mixture strategy group was immunized with an equal mixture of HPISVpmv1 from both NF54 and FCR3. The total amount of VAR2CSA immunogen used was kept constant and divided equally between variants. (**B**) The antibody titers from the rats measured by ELISA. VAR2CSA NF54 was coated on the plate. Each point is the mean of three technical replicates for each individual rat, and the bar represents the median value for the group. The p-values were calculated using a two-sided Kruskal-Wallis non-parametric test with Dunn's multiple comparisons compared to control and comparisons with p-values less than or equal to 0.05 are shown. (**C**) The antibody titers from the rats measured by ELISA. VAR2CSA FCR3 was coated on the plate. Each point is the mean of three technical replicates for each individual rat, and the bar represents the median value for the group. The p-values were calculated using a two-sided Kruskal-Wallis non-parametric test with Dunn's multiple comparisons compared to control and comparisons with p-values less than or equal to 0.05 are shown. (**D**) BIA assay results against the NF54 parasite strain using purified pooled IgG at 1 mg/ml from the serum after two vaccinations. The dashed line indicates the 50% inhibition level used as a cutoff for inhibitory or non-inhibitory activity. The data shown are derived from two independent experiments, and each point is the mean of two technical replicates. (**E**) BIA assay results against the CS2 parasite strain that contains FCR3 VAR2CSA using purified pooled IgGs at 1 mg/ml from the serum after two vaccinations. The dashed line indicates the 50% inhibition level used as a cutoff for inhibitory or non-inhibitory activity. The data shown are derived from two independent experiments, and each point is the mean of two technical replicates. This figure was generated with the help of Biorender (https://www.biorender.com/) and PRISM 9.

examined whether a mixture or sequential immunization with HPISVpmv1 from the NF54 and FCR3 strains with a two-vaccination regimen formulated in AddaS03 increased the breadth of protection (Fig 4A).

Both the mixture and sequential strategies induced strong immune responses (Fig 4B and 4C). Strikingly, both strategies also induced strong functional inhibitory activity against NF54 and CS2 parasite strains (Fig 4D and 4E). The CS2 parasites have a VAR2CSA sequence identical to that found in FCR3. However, neither approach was able to increase the breadth of protection to the parasite strain WF12, which has a VAR2CSA sequence identical to that found in 7G8 (S6 Fig). These data suggest that a cocktail vaccination strategy based on the highly expressible and adaptable HPISVpmv1 design may adequately expand the protection breadth.

HPISVpmv1 from three additional strains, M. Camp, M200101 and 7G8 were further designed to expand the scope of study to five variants. High expression yields for all these variants were observed, demonstrating the successful application of the HPISVpmv1 design to multiple variants of VAR2CSA with diverse sequences (S7A and S7B Fig and S1 Table). The five variants were evaluated in a mixture cocktail strategy to observe changes in breadth. All combinations were evaluated in a two-vaccination regimen formulated in AddaS03 with a total combined dose of 10 μg divided equally among all variants used (Fig 5A). Therefore, the dose of each individual variant was lower in the groups containing a larger number of variants. We tested combinations of three strains (NF54, FCR3, M. Camp), four strains (NF54, FCR3, M. Camp, M200101) and five strains (NF54, FCR3, M. Camp, M200101, 7G8). This experimental design allowed evaluation of binding inhibitory activity against NF54 and CS2 (VAR2CSA FCR3) parasites as readouts for the response to antigens included during vaccination and binding inhibitory activity against WF12 (VAR2CSA 7G8 sequence) parasites as the readout for breadth outside the set of antigens immunized. All strategies induced strong immune responses (Fig 5B) and retained potent inhibitory activity against NF54 and FCR3 (Fig 5C) despite lowering the dose for these two variants in the combination strategies. Intriguingly, only the rats immunized with HPISVpmv1 7G8 in the five-combination group or the 7G8 alone group showed potent inhibitory activity against WF12 (VAR2CSA 7G8 sequence) (Fig 5C). Taken together, these data indicate that strain-specific responses appear dominant and that a cocktail vaccination strategy using the adaptable HPISVpmv1 can expand the protection breadth.

## Discussion

Women can become susceptible to malaria infection during pregnancy despite any residual immunity from past *P. falciparum* infections. Pregnant women may further act as a reservoir for parasites, complicating malaria eradication efforts. An effective vaccine to prevent placental malaria is urgently needed, as parasites continue to acquire medication resistance, and new drugs carry the risk of teratogenesis [1,60]. Despite the critical role of VAR2CSA in placental malaria, the full-length protein is not ideal for industrial production under GMP conditions due to its size and complexity [61,62]. It is of great interest to develop an effective immunogen suitable for production that can elicit potent functional activity.

Women living in malaria-endemic areas develop immunity to placental malaria after successive pregnancies [9,63–65] yet the mechanisms of protection are not fully elucidated. Preclinical animal studies using diverse VAR2CSA fragments have identified potential inhibitory epitopes in VAR2CSA, and inhibitory antibodies have been raised to all DBL domains [62]. It is likely that antibodies binding to the CSA-binding site will be a potent mechanism to block the CSA-binding ability of VAR2CSA, thus preventing the placental accumulation of the parasite. In addition, antibodies may facilitate other effector functions that contribute to protection, including antibody-facilitated opsonic phagocytosis of infected red cells [35,66]. The recently solved cryo-EM structures of the VAR2CSA-CSA complex revealed the specific binding mechanisms between VAR2CSA and CSA [11,12]. We hypothesized that a VAR2CSA

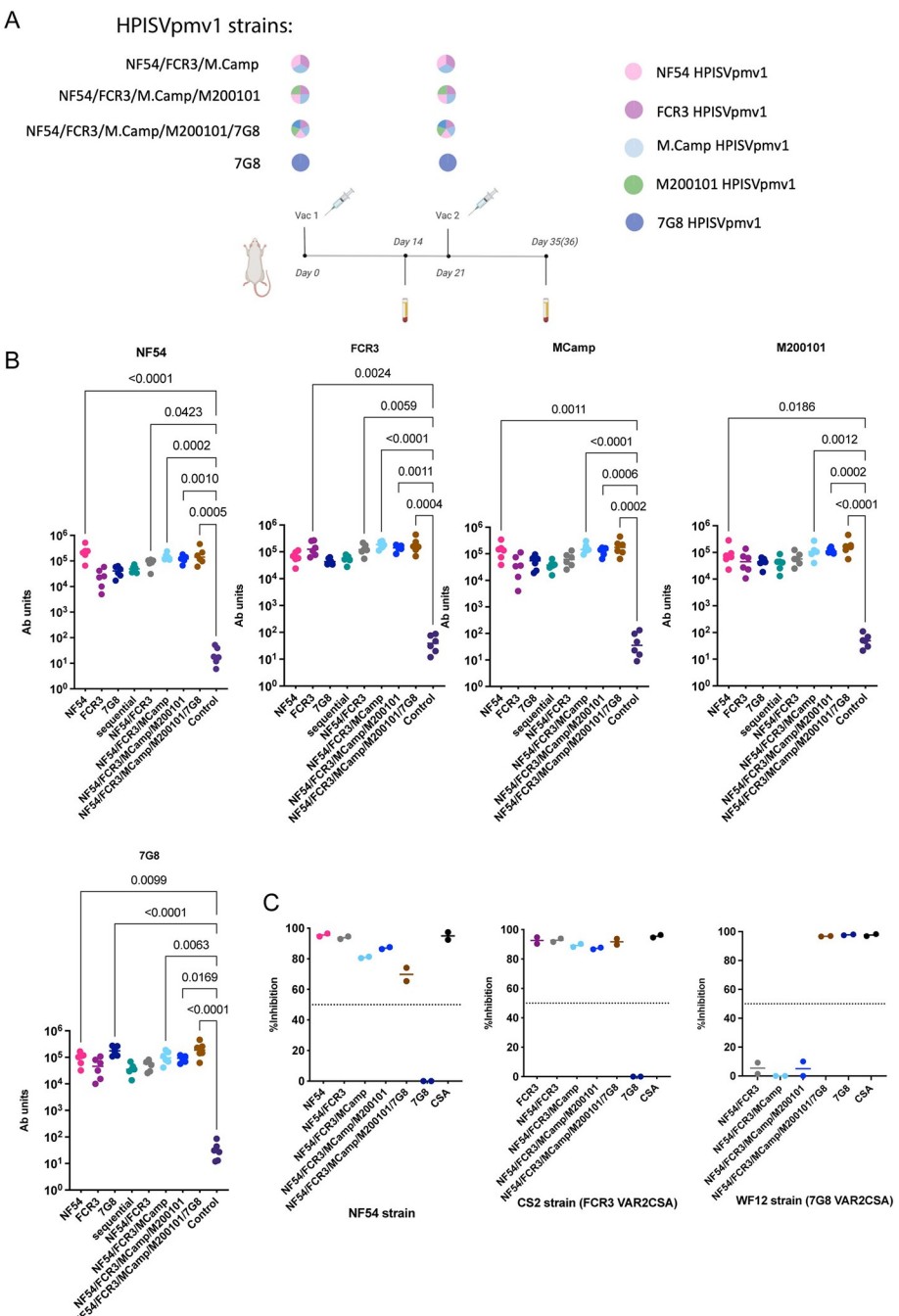

**Fig 5. Cocktail immunization with up to five HPISVpmv1 variants enhanced the breadth of inhibition.** (**A**) The two-vaccination immunization schedule with the cocktail strategy. The groups were immunized with different combinations of HPISVpmv1 variants from NF54, FCR3, M200101, M. Camp and 7G8. The HPISVpmv1 immunogens were equally combined to generate a mixture of three (NF54/FCR3/M. Camp), four (NF54/FCR3/M. Camp/M200101), and five (NF54/FCR3/M. Camp/M200101/7G8) strains for immunization. The total amount of VAR2CSA immunogen used was kept constant and divided equally among variants. (**B**) Antibody titers from the rats after each of the two immunizations. The HPISVpmv1 immunogens of each strain were coated on the plate as annotated above each graph. Each point is the mean of three technical replicates for each individual rat, and the bar represents the median value for the group. The cocktail combinations of HPISV strains are shown by the titles of the X-Axis. The p-values were calculated using a two-sided Kruskal-Wallis non-parametric test with Dunn's multiple comparisons compared to control and comparisons with p-values less than or equal to 0.05 are shown. (**C**) BIA assay results against NF54, CS2 (VAR2CSA FCR3 sequence), and WF12 (VAR2CSA 7G8 sequence) parasite strains using purified pooled IgGs at 1 mg/ml from the serum after two vaccinations. The dashed line indicates the 50% inhibition

level used as a cutoff for inhibitory or non-inhibitory activity. The cocktail combinations of HPISV strains are shown by the titles of the x-axis. The data shown are derived from two independent experiments, and each point is the mean of two technical replicates. This figure was generated with the help of Biorender (https://www.biorender.com/) and PRISM 9.

vaccine design that primarily focuses on the CSA binding sites within VAR2CSA will elicit potent inhibitory activity. Several immunogens were designed based on this hypothesis and shown to be as immunogenic and effective as the full-length protein in preclinical animal studies formulated with either CFA/IFA or AddaS03 in a two-vaccination regimen.

HPISVpmv1 has multiple advantages over current vaccine candidates and the full-length protein for placental malaria vaccine design. First, HPISVpmv1 is highly stable and expresses at six times the yield of the full-length protein. Second, HPISVpmv1 contains the complete CSA-binding site and inhibitory epitopes on DBL4, which are absent in PRIMVAC and PAM-VAC. Previous studies have identified inhibitory epitopes on DBL4, highlighting the importance of including this domain in any effective VAR2CSA vaccine. This domain, and the complete CSA binding site are included in HPISVpmv1 but are not present in other leading vaccine candidates. Third, we have shown that HPISVpmv1 is highly adaptable, producing high protein yields for all five variants tested here. This has major benefits from a manufacturing standpoint where protein yields, stability and adaptability to new variants are key advantages. Higher expression levels can increase yields during manufacturing of protein-based vaccines, thereby reducing costs. Expression level is also a key factor for mRNA vaccines, where the protein antigen must be expressed from a coding mRNA. The higher the expression level, the better the immunogenicity of mRNA vaccines. The expression level becomes further important once multiple variants are introduced in a single mRNA vaccine, as the total mRNA dose must be kept constant and distributed across multiple variant mRNAs. Higher expression levels can compensate for the loss of mRNA dose per variant. Fourth, HPISVpmv1 has the advantage of a larger mass than existing vaccines, and the antigen mass can enhance immunogenicity. This is especially pertinent in the context of cocktail immunizations, as the total vaccine dose is distributed among multiple variants. We have also demonstrated that HPISVpmv1 is highly immunogenic even at low doses required for immunization with multiple variants.

The structural studies indicated that the arm comprised of DBL5 and DBL6 does not play a role in CSA binding. Consistent with this finding, immunization with the arm alone generated high antibody titers comparable to the full-length ectodomain of VAR2CSA and HPISVpmv1 but with no functional binding inhibition activity. These results suggest that DBL5 and DBL6 may divert the immune response away from the core functional elements of VAR2CSA. However, epitopes in the arm correlated with functional activity have been reported [49,50,67,68] and the mechanism of protection for these epitopes remains to be further investigated.

Although ID1 is not seen in the full-length VAR2CSA structures [11–13], a recent study suggests that it may contact DBL2 and DBL4, contributing to the stabilization of the CSA binding channel [48]. In addition, although recombinant ID1 alone does not induce a sufficient protective antibody response, it appears indispensable for the induction of adhesion-inhibitory antibodies by either NTS-DBL1 or DBL2 [69]. Deletion of ID1 described here reduced the stability of multiple distinct designs and their ability to elicit an inhibitory response. This is further evidence that advocates for the inclusion of ID1 as a component of VAR2CSA immunogens.

A key tenet for a placental malaria vaccine is to develop immunity to diverse parasite isolates given the high variation in VAR2CSA. The data presented here indicate that

immunization with the full-length ectodomain of VAR2CSA results in inhibition that is limited to the strain immunized, referred to as homologous protection. This is consistent with previous findings that full-length VAR2CSA can induce broadly reactive antibodies to various strains with no strain transcendent inhibition [70]. A similar strain-specific response is also found for PRIMVAC and PAMVAC [44,45]. These data suggest that a single strain of VAR2CSA may not be sufficient to confer the breadth of protection desired against multiple parasite isolates. This is consistent with previous studies demonstrating that VAR2CSA-purified IgG from multigravida women showed broad inhibitory activity [71]. However, the cross-inhibitory reactivity purified from one strain appears limited to a certain number of other strains [71]. Interestingly, although immunization of a certain strain alone (NF54, FCR3 or 7G8) induces antibodies that cross-react with the other strain, these cross-reactive antibodies are not able to mediate inhibitory activity.

The cocktail strategy presented in this study offers a potential approach for a broadly protective placental malaria vaccine and similar practices have been employed for the development of influenza, pneumococcal, and COVID vaccines [57–59]. We demonstrate that increasing the number of variants used enhanced the breadth of protection to additional parasite strains. Notably, these results were achieved with a fixed total dose divided equally among the tested variants. Therefore, in the five-combination group, each HPISVpmv1 variant immunogen induced potent inhibitory activity at a low 2 µg dose in rats. These data suggest that it is possible to further expand the number of variants and lower the dose without having a detrimental effect on the inhibitory activity. The mixture strategy was also designed to evaluate whether a response can be generated to parasite strains not included during immunization. Interestingly, the results showed that rats only generated inhibitory activity to the strains that were included during vaccination, and breadth outside the strains used was not generated.

Together, these results suggest that a successful placental malaria vaccine may require a cocktail strategy that includes a limited number of circulating strains in endemic areas with consideration of the clades that form the VAR2CSA variation [72]. Future bioinformatic and field surveys will help determine the number of strains to include in this strategy, and clinical trials will be required to determine the efficacy of these combinations. Moreover, strain-specific inhibitory epitope mapping may facilitate additional engineering to refocus the antibody response to shared inhibitory epitopes. The cocktail vaccines could be formed by multiple protein HPISVpmv1 variants formulated in a suitable adjuvant. mRNA vaccine technology has recently become an excellent platform for a number of diseases, and cocktail mRNA vaccines are readily developed and have shown excellent efficacy and breadth. The design of an adaptable, stable, high-expressing VAR2CSA antigen that enables a cocktail strategy to increase the breadth of protection will enable the development of next-generation placental malaria vaccines.

## Materials and methods

### Ethics statement

Animal studies were performed in an American Association for Accreditation of Laboratory Animal Care-accredited facility under the guidelines and approval of the Institutional Animal Care and Use Committee at the National Institutes of Health.

### Structure-based design of VAR2CSA immunogens

The rational designs of the novel VAR2CSA immunogens are depicted in Fig 1D. In brief, the full-length VAR2CSA protein was first separated into two parts: the Core and the Arm. By maintaining the CSA binding sites, the Core is further simplified by subtracting DBL3 to make

Major and minor 2 and then subtracting ID2 to make the Major and minor 1. For Core, Major and minor 2 and Major and minor 1, versions lacking ID1 (ΔID1) were created. Removal of domains was stabilized by a flexible GS linker reconnecting the remaining segments. The length and sequence of the flexible linker were determined by evaluating the distance between connecting residues in the structure and designing GS-linker sequences that could accommodate those distances. The designed DNA sequences were codon-optimized and synthesized by GeneScript. The designed sequences are listed in S2 Table.

## Expression and purification of immunogens in Expi293 cells

The designed immunogens together with the wild-type VAR2CSA NF54 proteins were expressed in Expi293 (Thermo Fisher Scientific) cells according to the manufacturer's protocols. In brief, the cells were grown to $5.0 \times 10^6$ for three to four passages after thawing. The day before transfection, 500 ml of culture was seeded at a density of $2.5–3.0 \times 10^6$ cells ml$^{-1}$ in a 2-liter flask. On the day of transfection, cells were diluted back to $2.5–3 \times 10^6$ before transfection. The plasmid DNA was diluted with 25 ml of Opti-MEM I medium (Thermo Fisher Scientific) to a final concentration of 1 μg ml$^{-1}$.

Then, 1.4 ml of ExpiFectamine 293 Reagent (Thermo Fisher Scientific) was diluted with 25 ml Opti-MEM I medium, gently mixed and incubated at room temperature for 5 min. The diluted ExpiFectamine 293 Reagent was then added to the diluted plasmid DNA, mixed by swirling, and incubated at room temperature for 20 min. The mixture was added to the cells slowly while swirling the flask. The flask was returned to the incubator at 37˚C and in 8% $CO_2$. After 20 h of incubation, ExpiFectamine 293 Transfection Enhancer 1 (Thermo Fisher Scientific) and ExpiFectamine 293 Transfection Enhancer 2 (Thermo Fisher Scientific) were added to the transfection flask.

The cultures were centrifuged at 5,000 revolutions per min for 15 min 5 d post-transfection. The supernatant was collected and loaded onto Ni Sepharose Excel columns (GE Healthcare), which were manually packed in a glass gravity column. The column was washed twice with 10 column volumes of wash buffer (25 mM 4-(2-hydroxyethyl)-1-piperazine-ethanesulfonic acid (HEPES), pH 7.4, 150 mM NaCl, 25 mM imidazole) and eluted with 5 column volumes of elution buffer (25 mM Tris-HCl, pH 7.4, 150 mM NaCl, 250 mM imidazole). The elutes were concentrated with a 100-kDa cutoff centrifugal filter unit (Millipore Sigma) to 1 ml and further purified by size-exclusion chromatography (Superose 6 Increase 10/300, GE Healthcare) in PBS buffer. The peak fractions were collected and verified by sodium dodecylsulfate–polyacrylamide gel electrophoresis before animal study. The protein yield was measured with three technical replicates from three biological replicates.

## Circular Dichroism

CD spectra were collected on a JASCO J-815 spectropolarimeter at 25 degrees C in a 1 mm path length quartz cuvette between 200 and 250 nm with the buffer background subtracted. Each CD spectrum is the average of five scans at 50 nm/min with a 1 nm bandwidth and a time constant of 0.5 seconds. The analysis and plotting of the spectra were performed with CAPITO [73] that utilizes the Protein Circular Dichroism Data Bank (PCDDB) [74].

## Rat immunizations

**Formulation for vaccination.** CFA/IFA: A dose of 10 μg of antigen in a volume of 50 μL of the immunogens mentioned in Fig 1D in DPBS was added to 50 μL of Complete Freund's Adjuvant (CFA) (Sigma Aldrich) on day 0 or Incomplete Freund's Adjuvant (IFA) (Sigma Aldrich) on days 21 and 42 and incubated with vortexing for 15 min at room temperature. The

control immunization was 50 μL DPBS added to 50 μL of CFA on day 0 or IFA on days 21 and 42 and incubated with vortexing for 15 min at room temperature. All formulations were kept at room temperature between formulation and injection into rats.

AddaS03: For single antigen formulations, a dose of 10 μg in a volume of 50 μL of immunogens in DPBS was added to 50 μL of AddaS03 (Invivogen) and mixed by pipetting up and down 10 times. For multi-antigen mixed formulations, a total dose of 10 μg antigen composed of equal amounts of the antigens in a total volume of 50 μL in DPBS was added to 50 μL of AddaS03 and mixed by pipetting up and down 10 times. For 2 antigens, the dose was 5 μg each. For 3 antigens, the dose was 3.3 μg each. For 4 antigens, the dose was 2.5 μg each. For 5, antigens the dose was 2 μg each. The control immunization was 50 μL DPBS added to 50 μL of AddaS03 and mixed by pipetting up and down 10 times. All formulations were kept at room temperature between formulation and injection into rats.

**Immunization.** Female CD rats approximately 7 weeks in age were obtained from Charles River Laboratories. On either day 0, 21, and 42 for 3 vaccination studies or day 0 and 21 for 2 vaccination studies, rats were immunized subcutaneously in one site in the inguinal region with a volume of 100 μL containing a dose of 10 μg of total antigen in either CFA/IFA or AddaS03 formulated as described above. For the sequential cocktail immunization assay, rats were first immunized with HPISVpmv1 NF54 formulated as described above on day 0 and then immunized with HPISVpmv1 FCR3 formulated as described above on day 21. For the mixture cocktail immunizations, rats were immunized with a total antigen dose of 10 μg of mixed antigen formulated as described above on days 0 and 21.

**Sample collection.** Rats were bled for sera on days 14 and 35/36 for the two-vaccination protocol and on days 14, 35, and 63 for the three-vaccination protocol.

## Generation of a standard curve for ELISA antibody unit measurement

Pooled serum from rats immunized with full-length VAR2CSA or HPISVpmv1 was used as a standard curve on each plate to calculate the antibody titers of individual animals in all groups. One antibody unit (AU) was defined as the dilution of the standard serum required to achieve an Abs450 value of 1. Each plate included triplicate 2-fold serial dilutions of the standard serum from 20 to 0.01 AU. Serum from each animal was diluted such that the Abs450 fell in the informative portion of the standard curve between 0.1 and 2.0. The Abs450 values for the standard curve were fit to a 4-parameter logistic curve, which was used to convert Abs450 values to AU for each individual animal. AU values for each individual animal were measured in triplicate on separate plates, and the average is reported.

## Antibody titer measurements

Nunc MaxiSorp plates (Thermo Fisher Scientific) were coated with 100 μl 0.02 mg/mL purified full-length VAR2CSA or HPISVpmv1 diluted in 50 mM Na-carbonate pH 9.5. Plates were coated overnight at 4˚C and then washed three times with PBST. Plates were blocked for 1 hour at room temperature with 2% BSA in PBST and then washed three times with PBST. A 100 μl series of diluted serum (1:10,000; 1:100,000; 1:1,000,000) was added to each well. The primary antibody was incubated for 1 hour at room temperature, the plates were washed three times with PBST, and 200 μl of 1:5000 peroxidase-conjugated anti-rat IgG was added (Jackson ImmunoResearch Laboratories, Inc. Cat. # 109-035-098). Plates were incubated for 30 minutes at room temperature and washed three times with PBST. Finally, 70 μl tetramethylbenzidine (TMB) (MilliporeSigma) was added and incubated for 10 minutes at room temperature before quenching with 70 μl 2 M $H_2SO_4$. Absorbance at 450 nm was measured using a Biotek Synergy H1 plate reader.

### Rat antibody purification

Rat serum was diluted 1:2 with Pierce Protein G IgG Binding Buffer and then incubated with protein G resin (Gold Biotechnology) for 1 h at room temperature. After washing the resin with Pierce Protein G IgG Binding Buffer, the rat IgG was eluted with Pierce IgG Elution Buffer (0.15 ml of Tris pH 9.0 buffer per ml of purified IgG was added to neutralize the eluted fractions). The purified IgG was then buffer exchanged with PBS.

### Parasite culture and purification

Laboratory-adapted parasite lines were synchronized to reach the trophozoite/schizont stages with a parasitemia >1%, and parasites were enriched using a gelatin gradient. Parasitemia was determined by Giemsa staining. Pelleted blood was resuspended in RPMI 1640 to 50% hematocrit. Then, two volumes of 1% gelatin RPMI solution were added and incubated for 30 min in a 37˚C water bath. The upper layer was transferred to a new tube, and the parasites were pelleted and washed three times with RPMI 1640 medium. The parasites were then used for a binding inhibition assay (BIA). A strain of *Plasmodium falciparum* parasite culture was genotyped by sequencing fragments of the DBL4 domain of the *var2csa* and *clag2* genes. Briefly, using the NF54 parasite sequence, primers were designed to amplify the DBL4 domain (Fw: 5'-AGATATTATTAAAGGCAACGATTTAGTGC and Rv: 5'-ATTTGTCCATTCTTCCAAC CATCT) and CLAG2 (Fw: 5'- TTAAGTCTTTTGTGTGAATACCAAGCA and Rv: 5'-ATA GGTGCATCAGATTTCCAATTAAAG). The expected amplicon sizes were 355 bp and 248 bp for DBL4 and CLAG2, respectively. For each parasite culture, DNA was extracted using the QIAamp DNA Mini Kit (Qiagen) following the manufacturer's recommendation. PCR of DBL4 and CALG2 was performed using Q5 Hot Start High-Fidelity 2X Master Mix (New England Biolabs) following the manufacturer's suggested protocol and an annealing temperature of 64˚C. PCR products were cleaned and concentrated using a Monarch PCR & DNA Cleanup Kit (New England Biolabs) visualized on an ethidium bromide gel to confirm band size and singularity and sent to PooChon Scientific for Sanger sequencing. The resulting sequences were then aligned to the reference parasite for sequence identity assessment.

### Binding inhibition assay (BIA)

The CSA binding inhibition assay was evaluated in a static binding inhibition assay using immobilized CSA as previously described [75]. In brief, CSA (Sigma) at 10 μg/ml in PBS was coated as 15 μl spots on a 100 x 15 mm Petri dish (Falcon 351029) by overnight incubation at 4˚C. The spots were blocked with 3% BSA in 1x PBS at 37˚C for 30 min. Before the binding assay, enriched mature trophozoite/schizont stages of infected erythrocytes (IEs) from different P. falciparum strains were diluted to 20% parasite density at 0.5% hematocrit, blocked with 3% BSA in complete RPMI medium and then incubated with total rat IgG for 30 min at 37˚C. IE suspended in IgG solution were then allowed to bind duplicated receptor spots for 30 min at 37˚C. Unbound IE were washed away, and bound IE were fixed, stained and quantified by microscopy. The percentage of inhibition was calculated relative to the wells containing IE without IgG. The negative percentage was set to 0.

## Supporting information

**S1 Fig. Domain structures of the designed immunogens.** Each immunogen is illustrated as a schematic representation and the sequences of the linker was shown. This figure was generated with the help of Biorender (https://www.biorender.com/) and PRISM 9.
(TIFF)

**S2 Fig. Purification of the designed immunogens.** (A) The size-exclusion chromatography (SEC) profile overlay of the designed immunogens after purification are shown. The yield of each immunogen is illustrated within parenthesis behind the names. The SEC profile of each immunogen was shown below. The results shown here are one representative purification with all purifications performed at least three times with consistent results. (B) SDS-PAGE analysis of the designed immunogens.
(TIFF)

**S3 Fig. Circular dichroism (CD) analysis of the designed immunogens.** (A) CD spectra of the immunogens. (B) Upper panel shows the CD spectra for the different immunogens as color-coded in the inset. The lower panel shows the MRE (Mean-residue-ellipticity) values at lambda = 222 nm versus lambda = 200 nm that indicate the folding state of each immunogen as color-coded in the inset. PCDDB—Protein Circular Dichroism Data Bank.
(TIFF)

**S4 Fig. Optimization of the HPISVpmv1 construct.** (**A**) Domain structure of the HPISVpmv1 domain structure. (**B**) SDS-PAGE analysis result of HPISVpmv1 and HPISVpmv1_ext from NF54 and FCR3 strains. 'ext' is short for extension. (**C**) The yield of each immunogen is determined using eight biological replicates. The p values of the one-way ANOVA test are shown. This figure was generated with the help of Biorender (https://www.biorender.com/).
(TIFF)

**S5 Fig. Neither full length VAR2CSA nor designs show strain transcending activity.** (**A**) The BIA assay results against WF12 parasite strain containing 7G8 VAR2CSA using purified pooled IgG at 1mg/ml from the serum after three vaccinations. (**B**) The BIA assay results against CS2 parasite strain containing FCR3 VAR2CSA using purified pooled IgG at 4mg/ml from the serum after three vaccinations. The dashed line indicates the 50% inhibition level used as a cutoff for inhibitory or non-inhibitory activity, the data shown are derived from two independent experiments and each point is the mean of two technical replicates.
(TIFF)

**S6 Fig. The cocktail strategies show limited strain transcending activity.** The BIA assay results against WF12 parasite strain containing 7G8 VAR2CSA using purified pooled IgG at 1mg/ml from the serum after two vaccinations. The dashed line indicates the 50% inhibition level used as a cutoff for inhibitory or non-inhibitory activity, the data shown are derived from two independent experiments and each point is the mean of two technical replicates.
(TIFF)

**S7 Fig. Purification of HPISVpmv1 from diverse strains.** The Size-exclusion chromatography (SEC) profile (**A**) and SDS PAGE gel (**B**) of HPISVpmv1 from 7G8, M.Camp and M200101 are shown.
(TIFF)

**S1 Table. Sequence identity among 5 HPISVpmv1 strains.** The sequence identity (%) between each of the two HPISVpmv1 strains are shown in the table.
(DOCX)

**S2 Table. Protein sequences of the designed immunogens.**
(DOCX)

## Acknowledgments

We thank Tom Wellems for the WF12 parasite line. We thank Palak Patel and Dashuang Shi for experimental assistance and J. Patrick Gorres and David Garboczi for assistance in editing the manuscript. This work was supported by the Intramural Research Program of the Division of Intramural Research, National Institute of Allergy and Infectious Diseases (NIAID), National Institutes of Health (NIH) to N.H.T. The funders had no role in study design, data collection and analysis, decision to publish, or preparation of the manuscript.

## Author Contributions

**Conceptualization:** Rui Ma, Niraj H. Tolia.

**Data curation:** Rui Ma.

**Formal analysis:** Rui Ma, Niraj H. Tolia.

**Funding acquisition:** Niraj H. Tolia.

**Investigation:** Rui Ma, Nichole D. Salinas, Sachy Orr-Gonzalez, Brandi Richardson, Tarik Ouahes, Holly Torano, Bethany J. Jenkins, Jillian Neal, Junhui Duan, Robert D. Morrison, Apostolos G. Gittis, Justin Y. A. Doritchamou, Niraj H. Tolia.

**Methodology:** Rui Ma, Nichole D. Salinas, Thayne H. Dickey, Niraj H. Tolia.

**Project administration:** Rui Ma, Irfan Zaidi, Lynn E. Lambert, Patrick E. Duffy, Niraj H. Tolia.

**Resources:** Niraj H. Tolia.

**Supervision:** Lynn E. Lambert, Patrick E. Duffy, Niraj H. Tolia.

**Validation:** Rui Ma, Niraj H. Tolia.

**Visualization:** Niraj H. Tolia.

**Writing – original draft:** Rui Ma, Niraj H. Tolia.

**Writing – review & editing:** Rui Ma, Nichole D. Salinas, Sachy Orr-Gonzalez, Brandi Richardson, Tarik Ouahes, Holly Torano, Bethany J. Jenkins, Thayne H. Dickey, Jillian Neal, Junhui Duan, Robert D. Morrison, Justin Y. A. Doritchamou, Irfan Zaidi, Lynn E. Lambert, Patrick E. Duffy, Niraj H. Tolia.

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
