## [Decision Letter · Decision Letter 0]

7 Sep 2023

Dear Dr. Tolia,

Thank you very much for submitting your manuscript "Structure-based design of VAR2CSA and a cocktail strategy for a placental malaria vaccine" for consideration at PLOS Pathogens. As with all papers reviewed by the journal, your manuscript was reviewed by members of the editorial board and by several independent reviewers. In light of the reviews (below this email), we would like to invite the resubmission of a significantly-revised version that takes into account the reviewers' comments.

1. As suggested by Reviewer 1, more description is needed of the design process and the validation of the immunogens. More information should be provided on how linker lengths were chosen and any other in silico considerations that went into the truncation designs. Additionally, data should be provided that the immunogens folded correctly (secondary structure and protein binding to CSA).

2. The head-to-head serological comparison of full-length VAR2CSA and HPISVpmv1 in figure 3F needs to be titrated below 0.1 mg/ml.

3. The discussion gives the impression that a cocktail vaccine is a practical approach to achieve strain-transcendence coverage. However, the cocktails did not broaden reactivity beyond the included immunogens. Given that extreme VAR2CSA polymorphism is raised as a major challenge to vaccine development, the authors should acknowledge that the present data suggests these immunogens elicit highly strain-specific inhibitory antibodies and they have not estimated the parasite population diversity of these strain-specific inhibitory epitopes. Therefore, it is difficult to gauge how easy or difficult it will be to design a broad coverage VAR2CSA cocktail. It may be necessary for additional engineering to refocus the antibody response on more shared inhibitory epitopes.

We cannot make any decision about publication until we have seen the revised manuscript and your response to the reviewers' comments. Your revised manuscript is also likely to be sent to reviewers for further evaluation.

Sincerely,

Joe Smith

Academic Editor

PLOS Pathogens

James Collins III

Section Editor

PLOS Pathogens

Kasturi Haldar

Editor-in-Chief

PLOS Pathogens

orcid.org/0000-0001-5065-158X

Michael Malim

Editor-in-Chief

PLOS Pathogens

orcid.org/0000-0002-7699-2064

1. As suggested by Reviewer 1, more description is needed of the design process and the validation of the immunogens. More information should be provided on how linker lengths were chosen and any other in silico considerations that went into the truncation designs. Additionally, data should be provided that the immunogens folded correctly (secondary structure and protein binding to CSA).

2. The head-to-head serological comparison of full-length VAR2CSA and HPISVpmv1 in figure 3F needs to be titrated below 0.1 mg/ml.

3. The discussion gives the impression that a cocktail vaccine is a practical approach to achieve strain-transcendence coverage. However, the cocktails did not broaden reactivity beyond the included immunogens. The authors should acknowledge that they have no way of estimating the parasite population diversity of these strain-specific inhibitory epitopes and therefore it is difficult to gauge how to design a broad coverage VAR2CSA vaccine. It may be necessary for additional engineering to refocus the antibody response on more shared inhibitory epitopes.

Reviewer's Responses to Questions

**Part I - Summary**

Reviewer #1: The manuscript by Ma et al uses the recent structures of the important pregnancy-associated malaria vaccine candidate VAR2CSA to guide the design of new version of this protein which are more effective as vaccine immunogens. The structures provide detailed insight into the architecture of this complex protein and the location of the ligand binding site, making it timely to take such an approach. Indeed, a complex antigen such as VAR2CSA would benefit from this as it is difficult to make, making a more readily produced antigen desirable.

The approach used is a simple one of just truncating regions of the protein, sometimes replacing domains with linkers. Therefore, while this is “structure-guided”, it does not use the most recent techniques. There is also little data presented to show that these truncations are properly folded, with the authors relying solely on mobility on size exclusion columns and a gel.

The authors next test these immunogens by rat immunisation and assessing ELISA signal and the ability of raised sera to block pRBCs from binding to CSA. The main success from the design approach was to generate an immunogen which is produced at higher yield. However, none of the immunogens generated a more effective inhibitory response than full-length VAR2CSA and most (perhaps all?) were less effective. This is interesting as these truncated immunogens were still very large, containing most of the native protein. Why was the quality of the generated antibodies (i.e. the inhibitory activity as a factor of total IgG) not better if only epitopes for non-inhibitory antibodies had been removed?

Finally, the authors show that their immunogen is very variant specific, but also show that immunisation with a ‘cocktail’ of different VAR2CSA variants can lead to a better cross-inhibitory response. This is nice data, although it is not clear that it is related to the ‘structure-based’ design part as the authors do not show whether this only happens with their design or would also occur with full-length VAR2CSA.

In summary, this is a worthwhile study, but there are issues with validation of the designed immunogens and sadly, it appears as though the immunogens generated are not an improvement in terms of immunogenicity, while one does express substantially better. The cocktail approach, which would probably work with full-length VAR2CSA is interesting. There is still work to do to make the best immunogen, as these designs are only a bit shorter than full length are do seem to give slighted lower inhibitory effects.

Reviewer #2: This review is regarding the paper Structure-based design of VAR2CSA and a cocktail strategy for a placental malaria vaccine by Ma and colleagues.

This paper describes the design of various potential vaccine candidates (mainly subunit vaccines) based on knowledge of the VAR2CSA antigens' biological functions and structure . These candidate vaccine antigens were then expressed and used to vaccinate rats. Antibody levels and function were measured. Due to issues in the past of a lack of heterologous immunity being elicited the authors also investigated the possibility of blended or sequential vaccination with antigens from different variants. Antibodies generated using vaccines from multiple variants recognized multiple parasite strains suggesting that a blended vaccine is a plausible approach.

A vaccine against placental malaria would help reduce disease. Hurdles to a vaccine include the large size of the antigen which can make its expression in-vitro difficult as well as a lack of immunity generated to heterologous strains in past sub unit vaccines.

This paper provides potential solutions to both of these queries by showing that subunit vaccines made of antigens from multiple strains can elicit functional antibodies in rats.

This has not been previously shown experimentally and is a significant step forward in the field.

This paper is well written and the methods used are appropriate.

**Part II – Major Issues: Key Experiments Required for Acceptance**

Reviewer #1: The main concern is the lack of description of the approach used for design and lack of validation of the designed proteins. There is almost no description of the design process. Was it just truncation? When linkers were used, how were they designed? What surfaces on the proteins were exposed on truncation of neighbouring domains and were they fixed or resurfaced?

Also the only demonstration of protein quality is a gel and a SEC trace. Many of the peaks on the SEC trace are very small, making it impossible to see their shape and so at the very least a SEC trace with equivalent loading is essential. In my view, the authors should also present CD traces to show correct secondary structure, SEC-MALLS or the equivalent and demonstration that the proteins bound to CSA with the same affinity as the original. The major reason for concern is that the deltaID1 behaves poorly, while not containing any of the CSA binding residues. Perhaps this has not been designed properly to allow it to fold?

Secondly, sometimes the manuscript does not present a sufficient range of concentrations of sera to determine whether the designed immunogen is as good as the original. For example, Figure 3F only shows concentrations which give very high inhibition and it looks as though the design is slightly worse at the lowest concentration presented than VAR2CSA. Without repeats, stats and measurements at a range of lower concentrations, it is not possible to make a firm conclusion here. An IgG titration is required.

Indeed, stats are required throughout as they are missing for most of the data.

Reviewer #2: no major issues noted

**Part III – Minor Issues: Editorial and Data Presentation Modifications**

Reviewer #1: Through-out the manuscript, the authors use value judgements, with words like “much higher” (line 27), “highly” (line 103), “strong” (line 110), “potent” (line 139) etc etc. In my view, scientific reports should just state the facts in a quantitative way, without using such value judgements.

I also think that the title is wrong. VAR2CSA is not designed. VAR2CSA-based immunogens?

Line 37-38 – seems truncated?

Line 59 ‘the design of VAR2CSA’ seems strange phrasing as VAR2CSA is a natural protein. ‘Re-design’, ‘Design based on the structure of VAR2CSA’?

Figure 2 – please show individual data throughout, rather than just means. Shows states throughout.

Figure 3 – please show individual data throughout, rather than just means. Shows states throughout. For F, please titrate down further to IgG concentrations lower than 0.1 mg/ml.

Figure 4 – please show individual data throughout, rather than just means. Shows states throughout.

Figure 5 – please show individual data throughout, rather than just means. Shows states throughout.

Figure S2 – how was loading decided on this set of SEC traces? Some of the curves really can’t be seen at this y axis scale. Show on different scales or repeat with equal loading? What column? How many repeats?

Figure S3 – I would move this into a main figure as this is the major advantage of the new immunogen.

Figure S5 + 6 – please show individual data throughout, rather than just means. Shows states throughout.

Reviewer #2: Some minor editorial comments

Line 63- refer to figure 1B

Final paragraph in the Introduction- please be clear that this study is in Rats

Lines 106- expand abbreviation of the adjuvant CFA/IFA

lines 114-116, consider rephrasing this sentence as it is unclear and incorrect as currently written

paragraph starting 140- I needed to reread this multiple times to understand, please rephrase to it is clearer

line 221- specify which vaccines you are referring to

line 271- specify this is in rats

Figure 5- it is really hard to determine what the vaccine was and the antigen was being used in the assays. Please rearrange figure and rewrite figure legend

PLOS authors have the option to publish the peer review history of their article (what does this mean?). If published, this will include your full peer review and any attached files.

Reviewer #1: No

Reviewer #2: **Yes: **Elizabeth Aitken
---

## [Decision Letter · Decision Letter 1]

6 Nov 2023

Dear Dr. Tolia,

Thank you very much for submitting your manuscript "Structure-guided design of VAR2CSA-based immunogens and a cocktail strategy for a placental malaria vaccine" for consideration at PLOS Pathogens. As with all papers reviewed by the journal, your manuscript was reviewed by members of the editorial board and by several independent reviewers. The reviewers appreciated the attention to an important topic. Based on the reviews, we are likely to accept this manuscript for publication, providing that you modify the manuscript according to the review recommendations.

AE's comment: Please address the reviewer's comments to be more circumspect in some of your conclusions related to the new secondary structure analysis. As noted by the reviewer, similar secondary structure does not indicate folded structure.  I am not requiring additional experimentation. However, it is important to state other possible reasons for some of the immunogenic outcomes with the truncated proteins. 

Sincerely,

Joe Smith

Academic Editor

PLOS Pathogens

James Collins III

Section Editor

PLOS Pathogens

Kasturi Haldar

Editor-in-Chief

PLOS Pathogens

orcid.org/0000-0001-5065-158X

Michael Malim

Editor-in-Chief

PLOS Pathogens

orcid.org/0000-0002-7699-2064

AE's comment: Please address the reviewer's comments to be more circumspect in some of your conclusions related to the new secondary structure analysis. As noted by the reviewer, similar secondary structure does not indicate folded structure.  I am not requiring additional experimentation. However, it is important to state other possible reasons for some of the immunogenic outcomes with the truncated proteins. 

Reviewer Comments (if any, and for reference):

Reviewer's Responses to Questions

**Part I - Summary**

Reviewer #1: The manuscript by Ma et al is improved in this round and some of my suggestions have been accepted and acted on. In particular, the sequences of the immunogens are included, the design process is clear, data points and statistics are included on the graphs, value judgements have been replaced with numbers and the authors have made some CD measurements to indicate secondary structure of their immunogens. These changes all make this a publishable piece of work.

It does still have limitations. Using only CD to show that a protein is correctly folded doesn't quite work as this only shows secondary structure context. which is why I also suggested a functional assay, such as CSA binding. This might be why there are still some inconsistencies in the data. For example, lines 120-122 says that the immunogens are all correctly folded due to CD and then 136-138 says that some are not immunogenic due to being destabilised. These seem contradictory to me. Perhaps the CD is giving a misleading outcome as these immunogens have the correct secondary structure but an altered tertiary structure? Or perhaps it is a more interesting reason - such as because ID1 is required for generation of a neutralising response? We can't know from the data.

Despite this, I recommend publication. The removal of half of var2csa, making something which expresses 6-fold better and still has the same efficacy is a worthwhile step forward and should be published, even if the data doesn't really tell us why it is as good. I therefore recommend publication.

**Part II – Major Issues: Key Experiments Required for Acceptance**

Reviewer #1: none

**Part III – Minor Issues: Editorial and Data Presentation Modifications**

Reviewer #1: I think that the authors should be more circumspect in some of their conclusions. For example, making clear that CD alone only tells that there is secondary structure and giving more possible reasons for outcomes, such as the reason why ID1 removal leads to loss of activity. They state destabilisation but then only show data which indicates that it is folded. Perhaps thermal melt CD, which is easily done, and if they don't want to do this, just to at least state that there are other possible reasons for their observations.

PLOS authors have the option to publish the peer review history of their article (what does this mean?). If published, this will include your full peer review and any attached files.

Reviewer #1: No

Figure Files:

Data Requirements:

Reproducibility:

References:

---

## [Editor Report · Decision Letter 2]

29 Nov 2023

Dear Dr. Tolia,

We are pleased to inform you that your manuscript 'Structure-guided design of VAR2CSA-based immunogens and a cocktail strategy for a placental malaria vaccine' has been provisionally accepted for publication in PLOS Pathogens.

Best regards,

Joe Smith

Academic Editor

PLOS Pathogens

James Collins III

Section Editor

PLOS Pathogens

Kasturi Haldar

Editor-in-Chief

PLOS Pathogens

orcid.org/0000-0001-5065-158X

Michael Malim

Editor-in-Chief

PLOS Pathogens

orcid.org/0000-0002-7699-2064
---

## [Editor Report · Acceptance letter]

27 Feb 2024

Dear Dr. Tolia,

We are delighted to inform you that your manuscript, "Structure-guided design of VAR2CSA-based immunogens and a cocktail strategy for a placental malaria vaccine," has been formally accepted for publication in PLOS Pathogens.

Best regards,

Michael Malim

Editor-in-Chief

PLOS Pathogens

orcid.org/0000-0002-7699-2064